# Enhancement of hMSC In Vitro Proliferation by Surface Immobilization of a Heparin-Binding Peptide

**DOI:** 10.3390/molecules28083422

**Published:** 2023-04-13

**Authors:** Maura Cimino, Paula Parreira, Victoria Leiro, Aureliana Sousa, Raquel M. Gonçalves, Cristina C. Barrias, M. Cristina L. Martins

**Affiliations:** 1i3S—Instituto de Investigação e Inovação em Saúde, Rua Alfredo Allen, 208, 4200-135 Porto, Portugal; 2INEB—Instituto de Engenharia Biomédica, Rua Alfredo Allen, 208, 4200-135 Porto, Portugal; 3Faculdade de Engenharia, Departamento de Engenharia Metalúrgica e de Materiais, Universidade do Porto, Rua Dr. Roberto Frias, 4200-465 Porto, Portugal; 4ICBAS—Instituto de Ciências Biomédicas Abel Salazar, Universidade do Porto, Rua de Jorge Viterbo Ferreira, 228, 4050-313 Porto, Portugal

**Keywords:** human Mesenchymal Stem Cells, protein adsorption, surface modification, cell culture

## Abstract

The use of human Mesenchymal Stem Cells (hMSC) as therapeutic agents for advanced clinical therapies relies on their in vitro expansion. Over the last years, several efforts have been made to optimize hMSC culture protocols, namely by mimicking the cell physiological microenvironment, which strongly relies on signals provided by the extracellular matrix (ECM). ECM glycosaminoglycans, such as heparan-sulfate, sequester adhesive proteins and soluble growth factors at the cell membrane, orchestrating signaling pathways that control cell proliferation. Surfaces exposing the synthetic polypeptide poly(L-lysine, L-leucine) (pKL) have previously been shown to bind heparin from human plasma in a selective and concentration-dependent manner. To evaluate its effect on hMSC expansion, pKL was immobilized onto self-assembled monolayers (SAMs). The pKL-SAMs were able to bind heparin, fibronectin and other serum proteins, as demonstrated by quartz crystal microbalance with dissipation (QCM-D) studies. hMSC adhesion and proliferation were significantly increased in pKL-SAMs compared to controls, most probably related to increased heparin and fibronectin binding to pKL surfaces. This proof-of-concept study highlights the potential of pKL surfaces to improve hMSC in vitro expansion possible through selective heparin/serum protein binding at the cell-material interface.

## 1. Introduction

Human Mesenchymal Stem Cells (hMSC) are a heterogeneous set of adult stromal cells that can be isolated from different tissues. They have excellent capacities for regenerative medicine purposes due to their wide differentiation potential and immunomodulatory role [1]. Currently, more than 1400 clinical trials are ongoing worldwide, with very promising results [2,3,4,5]. However, independently of the tissue source, hMSC are scarce in the human body. Therefore, in vitro cell expansion before therapeutic applications is mandatory.

Cell behavior is a dynamic and complex process that depends on multiple signals from the cellular microenvironment, whose main component is the extracellular matrix (ECM). ECM is a 3D complex network of macromolecules (mainly glycosaminoglycans (GAGs), proteins and proteoglycans) where cells are organized in a specific way for each cell/tissue. The ECM structure and composition are highly dynamic, and the cell-matrix interactions directly control fundamental cellular functions, such as adhesion, migration, proliferation, and differentiation, among others [6]. ECM adhesive proteins (e.g., fibronectin, vitronectin, collagen) contain highly conserved domains that bind to cell surface integrin receptors. Moreover, GAGs, such as heparan sulfate, recruit heparin-binding molecules, including adhesive proteins and soluble growth factors, to the cell membrane, amplifying signal transduction [7,8,9,10,11]. Thus, one strategy to improve hMSC culture in vitro consists of using ECM components as inducers of adhesion and cell growth. Attachment-dependent cells, such as hMSC, are mediated in vitro by soluble growth factors present in fetal bovine serum (FBS)-supplemented medium and by cell adhesive proteins, such as fibronectin and vitronectin, that adsorb at the plates surface and change their conformation, exposing cell-binding domains [12,13]. We recently demonstrated that under serum-free conditions, hMSC growth is enhanced by medium supplementation with basic fibroblast growth factor (bFGF) and transforming growth factor β (TGF-β) [14]. It is also known that FGF uses heparan sulfate/heparin as a cofactor to bind to its cognate receptor (FGFR) [15,16,17]. Furthermore, heparin protects FGF from degradation, increasing the stability of the FGF-FGFR complex assembly and extracellular signal-regulated kinase (ERK) signaling dynamics [18].

Due to heparin’s properties in cell culture, this GAG has been incorporated into several biomaterials (reviewed in [19]) with the goal of attracting growth factors and enhancing their interactions with cell membrane receptors while avoiding their degradation. Nonetheless, the effect of heparin as a soluble cell culture supplement on hMSC behavior is controversial since both growth promotion [20] and suppression have been reported [21], which can be associated with the concentration of heparin used [22].

Surface immobilization of a small heparin-binding peptide together with an integrin-binding peptide enhanced adhesion and proliferation of hMSC in vitro by sequestration of serum-borne heparin to the cell-material interface [23]. We previously demonstrated that a synthetic heparin-binding polypeptide, composed of L-lysine and L-leucine (pKL), was able to bind heparin from human plasma after its immobilization onto model surfaces, namely self-assembled monolayers functionalized with tetra (ethyleneglycol) (EG4-SAMs) [24]. 

This work aims to evaluate if pKL-surfaces can be used as a new and more efficient support for the in vitro expansion of hMSC. It is expected that these surfaces will enhance hMSC proliferation by sequestration of serum-borne heparin, which will allow adhesive proteins and growth factors to congregate in a bioactive conformation at the cell-material interface.

## 2. Results

### 2.1. pKL Synthesis

The block co-polypeptide pKL, synthesized as described in Section 2.1, was characterized by infrared spectroscopy (FTIR). The obtained spectrum (Appendix A) was in accordance with the described in the literature [25]. As such, both monomers polymerized, giving rise to the copolymer pKL, consisting of 25 L-Lysines and 25 L-Leucines, with a molecular weight of around 8500 Da, in accordance with the literature [24]. 

### 2.2. pKL-SAMs

The success of pKL covalent immobilization on EG4-SAMs was confirmed by the increase in surface thickness and the water contact angle. 

Figure 1A shows the average thickness of the different surfaces determined by ellipsometry. In the absence of pKL, EG4-SAMs displayed a thickness ranging from 2.5 ± 0.3 nm (EG4-SAMs) to 3.5 ± 0.3 nm after CDI activation (30 CDI buffer). Following pKL immobilization, the average thickness ranged from 2.8 ± 0.3 nm when pKL was only adsorbed on EG4-SAMs (0 CDI pKL) to 7.6 ± 1.2 nm when pKL was immobilized on surfaces pre-activated with the higher CDI concentration (30 CDI pKL). These results are in accordance with those obtained in our previous work [24], confirming that pKL was covalently immobilized onto CDI-activated surfaces. 

Figure 1B depicts the water contact angle measurements for the different surfaces. Contact angle measurement is a qualitative way to assess surface wettability. It is based on the evaluation of the intermolecular interactions between the surface and a small drop of water when the drop meets the surface. By definition, water contact angles of hydrophilic surfaces are typically smaller than 90°, while hydrophobic ones have contact angles greater than 90° [26].

Concerning surface activation with CDI, the water contact angle of surfaces varied from 38 ± 2° for EG4-SAMs to 55 ± 4° after CDI activation (30 CDI buffer). In comparison, the water contact angle of pKL-SAMs ranged from 61 ± 1° when pKL was only adsorbed (0 CDI pKL) to 66 ± 3° when pKL was grafted on CDI-activated SAMs (30 CDI pKL). These results also suggest that pKL was successfully grafted onto CDI-activated EG4-SAMs. However, with this technique, some adsorbed pKL is also detected, as previously reported [24].

### 2.3. Heparin and Fibronectin Adsorption on pKL-SAMs

The ability of pKL-SAMs to bind heparin, fibronectin or fetal bovine serum (FBS) was evaluated using a Quartz Crystal Microbalance with Dissipation (QCM-D). Results in Figure 2 were calculated by applying the Voigt model to QCM-D data (frequency and dissipation). Representative QCM-D data plots are shown in Appendix A).

A significantly higher amount of heparin was adsorbed on pKL-SAMs when compared with the controls (0.3 and 30 CDI buffer) (Figure 2A). Moreover, twice the amount of heparin (379 ± 2 ng/cm^2^) adsorbed on 30 CDI pKL surfaces compared to that adsorbed on 0.3 CDI pKL (182 ± 0.01 ng/cm^2^). Heparin adsorption to the controls 0.3 and 30 CDI buffer was very low (31 ± 1 ng/cm^2^ and 27 ± 2 ng/cm^2^, respectively). These results demonstrated the capacity of pKL-SAMs to bind heparin in a concentration-dependent way.

Fibronectin adsorption to pKL surfaces (Figure 2B) also occurred in a concentration-dependent manner: 926 ± 67 ng/cm^2^ of fibronectin adsorbed to 30 CDI pKL surface and 693 ± 112 ng/cm^2^ to 0.3 CDI pKL surface. However, fibronectin adsorption was also detected on the control surfaces: 532 ± 65 ng/cm^2^ for 30 CDI buffer and 159 ± 39 ng/cm^2^ for 0.3 CDI buffer surfaces, suggesting that pKL also binds fibronectin in a non-specific way.

Protein adsorption on pKL surfaces incubated with 1% FBS was also measured by QCM-D (Figure 2C) to evaluate whether proteins present in serum-supplemented cell culture medium would also bind to the cell culture surface. The amount of serum proteins adsorbed was 774 ± 1 ng/cm^2^ on 30 CDI pKL and 281 ± 2 ng/cm^2^ on 0.3 CDI pKL surfaces, against 489 ± 7 ng/cm^2^ on 30 CDI buffer and 131 ± 32 ng/cm^2^ on 0.3 CDI buffer. Results suggested that, in comparison with fibronectin adsorption from a pure solution (Figure 2B), the decrease of adsorbed mass from FBS in 0.3 CDI pKL SAMs could be related to heparin adsorption. This fact was based on the described specific heparin adsorption on surfaces with lower pKL concentration, namely surfaces activated with CDI concentrations up to 0.3 mg/mL [24].

### 2.4. hMSC Adhesion and Morphology on pKL-SAMs

hMSC morphology on pKL-SAMs and TCPE surfaces was evaluated by fluorescent staining of F-actin, vinculin and nuclei 24 h after cell seeding (Figure 3A). hMSC on pKL-SAMs (0.3 CDI- and 30 CDI-pKL) showed a more elongated shape (in green) and more focal adhesions formation (in red), suggesting a better cytoskeleton organization than cells on TCPE or in SAMs with adsorbed pKL (0 CDI). Metabolic activity evaluation on day 1 (Figure 3B) suggested that pKL surfaces promoted hMSC adhesion since adhesion was significantly increased on pKL surfaces (30 CDI-pKL) as compared to TCPE.

### 2.5. hMSC Metabolic Activity/Growth on pKL-SAMs

hMSC metabolic activity on pKL-SAMs and controls (buffer-SAMs) was analyzed to infer cell growth (during 7 days) (Figure 4).

Cells’ metabolic activity was significantly higher in pKL-SAMs (either 0.3 or 30 CDI pKL) than in surfaces where pKL was only adsorbed (0 CDI pKL) and in the controls (30 CDI buffer). Moreover, all the pKL-SAMs had a better hMSC growth-promoting effect than TCPE surfaces. Statistically significant differences were only observed after 5 days in culture (Figure 4).

## 3. Discussion

In the present work, a heparin-binding synthetic polypeptide (poly(L-lysine, L-leucine) (pKL)) was immobilized on self-assembled monolayers (SAMs) to evaluate its potential to improve hMSC in vitro expansion. SAMs were used since they are easy to produce, functionalize and characterize at the molecular level [27]. Moreover, SAMs are relatively stable in biological studies when protected from oxidation, which can be achieved by performing the assays shortly after SAMs preparation. Additionally, in the presence of serum, the adsorbed protein layer can protect SAMs from oxidation during cell adhesion assays and through the first days of cell proliferation [28]. pKL was immobilized onto tetra (ethylene glycol) terminated SAMs (EG4-SAMs) due to its non-fouling characteristics that strongly reduce nonspecific adsorption [29]. Earlier, it was demonstrated that pKL immobilized onto EG4-SAMs in low concentrations bound heparin from human plasma in a selective way [24]. The increase in surface thickness and hydrophobicity after pKL immobilization onto CDI pre-activated SAMs demonstrated that pKL was successfully immobilized onto EG4-SAMs. More details on surface characterization, namely using X-ray photoelectron spectroscopy (XPS), Fourier transform infrared reflection absorption spectroscopy (IRAS), atomic force microscopy (AFM) and electrokinetic analyzer (EKA) techniques can be found in [24,30].

pKL-SAMs improved hMSC adhesion and proliferation. Metabolic activity, evaluated as an index of cell adhesion (24 h after seeding), was significantly increased on pKL-SAMs (30 CDI pKL) as compared with tissue culture polyethylene coverslips (TCPE). Moreover, upon 24 h, hMSC on pKL-SAMs (0.3 and 30 CDI pKL) exhibited a more elongated shape and a more defined cytoskeleton organization than TCPE. In addition, the increase in hMSC metabolic activity over seven days in culture, here used as an indication of cell growth, was much more significant on pKL-SAMs as compared to TCPE and to the other controls (0 CDI pKL and 30 CDI buffer). No differences among surfaces with pKL grafted in different concentrations (0.3 CDI and 30 CDI pKL) were observed throughout the assay (seven days).

The higher hMSC adhesion and proliferation on pKL surfaces could be related to heparin adsorption since it was previously demonstrated that heparin immobilized on anti-adhesive surfaces increased hMSC adhesion [31].

Heparin, fibronectin and FBS adsorption to pKL-SAMs was followed using Quartz Crystal Microbalance with Dissipation (QCM-D) assays. QCM-D is a powerful and high-sensitivity tool to quantify molecules (proteins, glycans and more) adsorption/binding to different substrates [27,29,32,33]. For data modeling, the Voigt model was selected. Although in some cases the simpler Sauerbrey equation could have been applied (buffer-SAMs; ΔD < 2 × 10^6^), surfaces with pKL elicited dissipation shifts that were compatible with a viscoelastic protein layer being formed (Appendix A). Additionally, using the Voigt model did not translate into significant changes in the overall mass (<1% change in modeled mass values) for surfaces where the Sauerbrey equation could have been applied.

Heparin-pKL surface recognition was herein demonstrated by QCM-D assays. The mass of heparin adsorbed was proportional to the amount of surface-bound peptide (higher heparin adsorption on 30 CDI pKL- than on 0.3 CDI pKL-SAMs). However, Martins et al. [24] stated that, in the presence of human plasma, surfaces with higher pKL concentration (30 CDI pKL) were not 100% selective for heparin, allowing the binding of other plasma proteins. Indeed, fibronectin adsorption to pKL-SAMs was very high, especially in 30 CDI pKL-SAMs. pKL-SAMs also adsorbed fetal bovine serum (FBS) in a pKL dose-dependent manner (higher FBS adsorption on 30 CDI pKL- than on 0.3 CDI pKL-SAMs). This may be related not only to the adsorption of heparin from FBS but also to the adsorption of other FBS proteins, such as albumin (the most abundant protein in FBS) or fibronectin (also present in large quantities [34]), mainly in 30 CDI pKL-SAMs that are not selective to heparin [24]. In addition, high FBS adsorption was also detected on 30 CDI buffer-SAMs. This might be related to the adsorption of other FBS proteins since very low heparin adsorption was observed on 30 CDI buffer-SAMs.

We then hypothesized that in hMSC standard growth medium (10% FBS), serum-borne heparin and other compounds, such as fibronectin, could adsorb at pKL-SAMs, exerting beneficial effects onto cell growth. Moreover, since fibronectin has multiple heparin-binding domains [35], it could be adsorbed onto pKL surfaces directly or through a heparin layer bound on pKL-SAMs. Importantly, when adsorbed through a heparin layer, fibronectin assumes the natural conformation found in the ECM [36], favoring cell adhesion. This could explain why both pKL surfaces (0.3 pKL and 30 CDI pKL) had a similar beneficial effect on hMSC growth since lower pKL concentrations could adsorb more heparin from a complex system [24]. On opposite, surfaces without pKL (0.3 and 30 CDI buffer) did not bind serum-borne heparin but could adsorb proteins. Adsorption of albumin, a non-fouling protein that hampers cell adhesion, could explain the lower index of hMSC growth on control surfaces regarding pKL surfaces. Importantly, this event was only detectable after five days in culture. During this period, cells may produce endogenous proteins and growth factors that are then adsorbed at the heparin layer on the surface. This, in turn, favors growth factors binding to cell receptors, amplifying the overall signaling cascade that generates cell proliferation. Further studies will investigate growth factor binding at the pKL surface in terms of identity and quantity.

Our data are in agreement with the literature when it was shown that hMSC proliferation was promoted in GAG-derivatized chitosan membranes containing heparin [37] or when a bioinert HS-EG3-SAMs was functionalized with a heparin-binding peptide (GGGKRTGQYKL) and an RGD peptide required for cell adhesion [23]. Nevertheless, the advantages of pKL as compared to other described heparin-binding peptides are (*i*) its relatively low production cost on a large scale (easy industry/clinical translation) and (*ii*) the immobilization simplicity that does not depend on its orientation on the surface.

As the main limitations, this study was fully conducted in model surfaces. In the future, it would be ideal to translate this strategy to polymer surfaces that can be used to coat tissue culture plates. Additionally, the differentiation potential and/or maintenance of the undifferentiated state of hMSC cultured on pKL surfaces was not addressed here and should be further investigated.

Finally, pKL surfaces could be combined with *xeno-free* media in hMSC expansion. The use of FBS is being banned due to its drawbacks, and chemically defined media and human plasma derivatives are being increasingly used [14,38]. However, the most general approach for hMSC *xeno-free* expansion requires surface coating with adhesive proteins and supra-physiological concentrations of growth factors in a cell culture medium, which results in high costs [14,39]. pKL surfaces may be used to overcome those limitations: pKL could interact with cell-surface glycosaminoglycans and amplify either fibronectin surface binding and cell-secreted growth factors signaling, resulting in a more affordable strategy.

Overall, this work is a proof-of-concept study presenting a surface-grafted synthetic heparin-binding polypeptide (pKL) able to be employed as a new substrate for hMSC culture, enhancing both cell adhesion and proliferation.

## 4. Materials and Methods

### 4.1. pKL Synthesis and Characterization

Poly(L-lysine, L-leucine) (pKL) was synthesized via ring-opening polymerization of the corresponding amino acid N-carboxyanydrides (NCAs), N-e-benzyloxycarbonyl-L-lysine-N-carboxyanhydride (Z-Lys NCA) and L-leucine-N-carboxyanhydride (Leu NCA), according to the experimental procedure described in [24]. pKL schematic representation and further synthesis details can be found in Appendix A).

pKL was characterized by Fourier Transform Infrared Reflection spectroscopy (FTIR) using a spectrophotometer (PerkinElmer, model 2000) (Appendix A). The pKL pellet was prepared by blending 2 mg of the co-polypeptide pKL with 200 mg of Potassium Bromide (KBr) previously dried at 105 °C for 24 h. The infrared spectra were obtained using 100 scans at 4 cm^−1^ spectral resolution. All spectra were recorded under dry nitrogen to eliminate the water vapor absorption.

### 4.2. pKL Surfaces Preparation and Characterization

#### 4.2.1. Gold Substrates

##### For Surface Characterization and Biological Assays

Gold-coated substrates (1 × 1 cm^2^), obtained from *Instituto de Engenharia de Sistemas e Computadores–Microsistemas e Nanotecnologias*, Portugal (INESC-MN), were prepared by the deposition of a thin layer of chromium (2.3 nm) and gold (37 nm) on silicon wafers as previously described [40]. Immediately before use, gold substrates were cleaned with fresh ‘‘piranha’’ solution [7 parts concentrated sulfuric acid (95 vol.%, BDH Prolabo, VWR- International), 3 parts hydrogen peroxide (30 vol.% Merck, Darmstadt, Germany)] for 5 min (caution: *this solution reacts violently with many organic materials and should be handled with care*). Gold-coated substrates were then rinsed sequentially with absolute ethanol (99.8%, Merck), Type II water (purified and deionized: Resistivity > 1 MΩ/cm; Conductivity < 1 µS/cm; Total organic carbons < 50 ppb) and absolute ethanol once again in an ultrasonic bath for 2 min, followed by drying with a gentle argon stream.

##### For Adsorption Assays Using Quartz Crystal Microbalance with Dissipation (QCM-D)

Gold-coated quartz crystal sensors (QSX301-Standard Gold; 78 mm^2^ active sensor area with a fundamental frequency of 5 MHz) were obtained from Biolin Scientific, Gothenburg, Sweden. Briefly, the sensors were cleaned for 10 min in a UV/ozone oven followed by a 5 min immersion in “piranha” solution as described above for gold-coated substrates.

#### 4.2.2. EG4-SAMs

EG4-SAMs were prepared as described elsewhere [24]. Briefly, (11-mercaptoundecyl)tetra(ethylene glycol) (EG4-thiol; SensoPath Technology, Seattle, WA, USA, SPT-0011P4) was diluted in absolute ethanol to a final concentration of 0.1 mM. Gold substrates (gold-coated substrates or gold-coated quartz crystal sensors) were immersed in EG4-thiol solutions and incubated for 24 h at room temperature (RT) in a nitrogen environment. Afterwards, formed self-assembled monolayers (EG4-SAMs) were rinsed three times in absolute ethanol and dried with an argon stream.

#### 4.2.3. pKL-SAMs

pKL immobilization onto EG4-SAMs proceeded via a two steps reaction (Appendix A): (a) activation of -OH groups of EG4-SAMs using N,N’-Carbonyldiimidazole (CDI; Sigma-Aldrich, St. Louis, MO, USA) to create reactive imidazolyl-carbamate groups (*CDI activation*); (b) pKL covalent immobilization on activated SAMs through its free NH_2_ groups (*pKL immobilization*).

To prepare EG4-SAMs with different degrees of activation, SAMs were immersed in solutions with different CDI concentrations: 0, 0.3 and 30 mg/mL, in anhydrous tetrahydrofuran (THF; Merck) as described elsewhere [24]. Then, CDI-activated EG4-SAMs were immersed in a pKL solution (0.5 mg/mL) prepared in dimethyl sulfoxide (DMSO; BDH Prolabo) with 0.02% (*v*/*v*) triethylamine (Sigma-Aldrich). Samples were incubated at 40 °C, 100 rotations per minute (rpm) agitation for 24 h. Control surfaces were prepared by incubating CDI-activated EG4-SAMs in a DMSO solution with 0.02% triethylamine (buffer-SAMs).

Afterwards, pKL-SAMs and buffer-SAMs were rinsed with DMSO (three times, during 5 min in an ultrasonic bath) and Type I water (Milli-Q^®^ ultrapure water: Resistivity > 18 MΩ/cm; Conductivity < 0.056 µS/cm; Total organic carbons < 50 ppb) (twice for 5 min in an ultrasonic bath). Finally, samples were dried with an argon stream and stored under an inert atmosphere until further use.

### 4.3. Surface Characterization

#### 4.3.1. Water Contact Angle

Water contact angle measurements were performed using a system from Data Physics, model OCA 15, equipped with a video CCD camera and SCA 20 software (DataPhysics Instruments GmbH, Filderstadt, Germany). SAMs were placed in a closed, thermostatic chamber saturated with water to prevent evaporation of the liquid from the drop. Measurements were performed using the sessile drop method with Type I water at 25 °C. After 4 μL drop deposition, images were taken every 2 s over 90 s, and the contact angle value was calculated at each time point according to the ellipse method. Then, the water contact angle for each surface was calculated by extrapolating the time-dependent curve to zero. Three replicates per sample were used.

#### 4.3.2. Ellipsometry

Ellipsometry measurements were performed with an ellipsometer model EP^3^ from Nanofilm Surface Analysis. The ellipsometer operates in a polarizer compensator-sample-analyzer (PCSA) mode (null ellipsometry). As a light source, a solid-state laser with a wavelength of 532 nm was used. The gold substrate refractive index (*n*) and extinction coefficient (*k*) were determined using a delta and psi spectrum with an angle variation between 66° and 76°. To correct any instrument misalignment, measurements were made in four zones. To determine the thickness of SAMs, the same kind of spectrum was used, and *n* and *k* for the organic layer were set as 1.45 and zero, respectively, as described [24]. Results are the average of eight measurements on each of the three samples for each type of SAM.

### 4.4. Heparin and Fibronectin Adsorption to pKL-SAMs

Quartz Crystal Microbalance with Dissipation (QCM-D) system (Q-Sense E4 instrument, Biolin Scientific, Gothenburg, Sweden) was used to monitor in real-time the frequency (∆f) and dissipation (∆D) shifts related to heparin, fibronectin or 1% (*v*/*v*) Mesenchymal Stem Cells certified fetal bovine serum (FBS) (ThermoFisher, Karlsruhe, Germany, Ref. 10094563) adsorption onto pKL-SAMs and controls (buffer-SAMs).

pKL-SAMs and buffer-SAMs assembled onto QCM-D gold-coated crystals were placed in the system, followed by injection (flow rate = 0.1 mL/min) of filtered (0.22 μm pore size) phosphate buffered saline solution (PBS, 0.1 M, pH 7.4) until a stable signal was obtained (baseline) following a protocol adapted from [33]. Then, either heparin sodium salt from porcine mucosa (0.1 mg/mL, Sigma-Aldrich ref. H3393), fibronectin from human plasma (40 μg/mL, Sigma-Aldrich Ref. F0895), or a 1% FBS solution in PBS were injected in the system (flow rate 0.1 mL/min). Incubation proceeded under static conditions at 37 °C for 2 h. Finally, rinsing was performed with PBS to remove any unbound protein (flow rate 0.1 mL/min).

Data were modeled using the Voigt model. This model is indicated when high dissipation changes (ΔD > 2 × 10^6^) are observed [41]. With the Voigt model, the soft/viscoelastic properties are taken into account for the quantification of a hydrated mass adsorbed on the surfaces [33]. Data from the 3rd to the 11th harmonics were collected and used in the analysis. The density and viscosity of the protein solutions were established at 1.35 g/cm^3^ and 0.0014 kg/ms (values commonly used for low-concentrated protein solutions) [42,43]. Results are presented in mass per area (ng/cm^2^) and represent the average of three independent assays with three replicates per sample.

### 4.5. Human Mesenchymal Stem Cells (hMSC) Adhesion and Proliferation on pKL-SAMs

#### 4.5.1. hMSC Culture

hMSC isolated from the bone marrow of a healthy 21-year-old black male (Lonza PT2501, VWR International, Radnor, PA, USA) were used. As these cells are commercially available, no patient consent or approval from Ethics Committee was required. Briefly, cells were cultured in Dulbecco’s Modified Eagle’s Medium (DMEM, Ref. 21885-108 Thermofisher Scientific) + 10% MSC qualified FBS (Thermofisher, ref. 12662029) + 1% Penicillin/Streptomycin (Thermofisher, ref. 15070063) at 37 °C in a 5% CO_2_ atmosphere. Cell culture media was changed twice a week, and cells were split by trypsinization (Sigma-Aldrich, T4799) when 70–80% confluence was reached. Cells between passages (P) 4 and 7 were used in this study.

#### 4.5.2. hMSC Cytoskeleton Organization

hMSCs were seeded on pKL surfaces and controls EG4-SAMs activated with different concentrations of CDI and tissue culture polyethylene coverslips (TCPE; 13 mm; Sarstedt, Numbrecht, Germany; ref. 83.1840.002) within 24-well suspension culture plates (Sarstedt; ref. 83.3922.500). After 24 h, cells were washed twice with pre-warmed PBS, fixed in 4% (*v*/*v*) paraformaldehyde for 15 min, incubated in permeabilizing buffer for 5 min (0.1% (*v*/*v*) Triton-X in PBS) and blocked with 1% (*w*/*v*) bovine serum albumin (BSA) in PBS at RT. To evaluate cytoskeleton organization, actin filaments were stained with Alexafluor-conjugated phalloidin (1:40 in 1% (*w*/*v*) BSA/PBS; Molecular Probes, A12379). To analyze focal adhesion formation, vinculin was immunostained with primary mouse anti-human monoclonal antibody for 1 h at 37 °C (hVIN-1, 1:100 in 1% (*w*/*v*) BSA/PBS, Sigma V-9131), followed by rabbit anti-mouse Alexa Fluor 594 conjugated Fab fragments IgG for 1 h at 37 °C (1:100 in 1% (*w*/*v*) BSA/PBS; Molecular Probes, A-21204). Nuclei were counterstained with 20 μg/mL 4,6-Diamidina-2-phenylin (DAPI; Sigma-Aldrich, D-9542) for 10 min at RT. Finally, samples were washed with PBS, mounted with Vectashield^®^ (Vector Labs, Newark, NJ, USA) in glass slides and photographed with a 20× objective of an inverted fluorescence microscope (Axiovert M100; Carl Zeiss, Oberkochen, Germany).

#### 4.5.3. hMSC Metabolic Activity

For metabolic activity evaluation, hMSC (P7) were plated at 3000 cells/cm^2^ in 24-well suspension culture plates (Sarstedt, Numbrecht, Germany; ref. 83.3922.500) in pKL surfaces and controls (EG4-SAMs activated with CDI and TCPE coverslips). Cells were cultured over 7 days in DMEM + 10% FBS + 1% P/S. At days 0 (4 h after seeding), 1, 3, 5 and 7, cell metabolic activity was assessed by resazurin assay as described elsewhere [44]. Assay was performed in triplicate.

### 4.6. Statistics

To determine the statistical significance of the results from the resazurin assay, a One-Way (from day 0 to day 1) and a Two-Way ANOVA (days 1 to 7) test for multiple comparisons were performed. For ellipsometry and contact angle measurements, a One-Way ANOVA test for multiple comparisons was performed. For QCM-D data, a T-test (non-parametric-Mann–Whitney test) was performed. Statistical significance was set for *p* < 0.05.

## 5. Conclusions

In this work, a new strategy for hMSC in vitro expansion was developed. It relies on the immobilization of a synthetic polypeptide (pKL) on model surfaces. The resulting surfaces were able to bind the most common ECM proteins (heparin and fibronectin), as described by QCM studies. hMSC adhesion and growth were significantly increased on pKL surfaces compared to controls, most probably through selective heparin/serum protein binding at the cell-material interface.

## Figures and Tables

**Figure 1 molecules-28-03422-f001:**
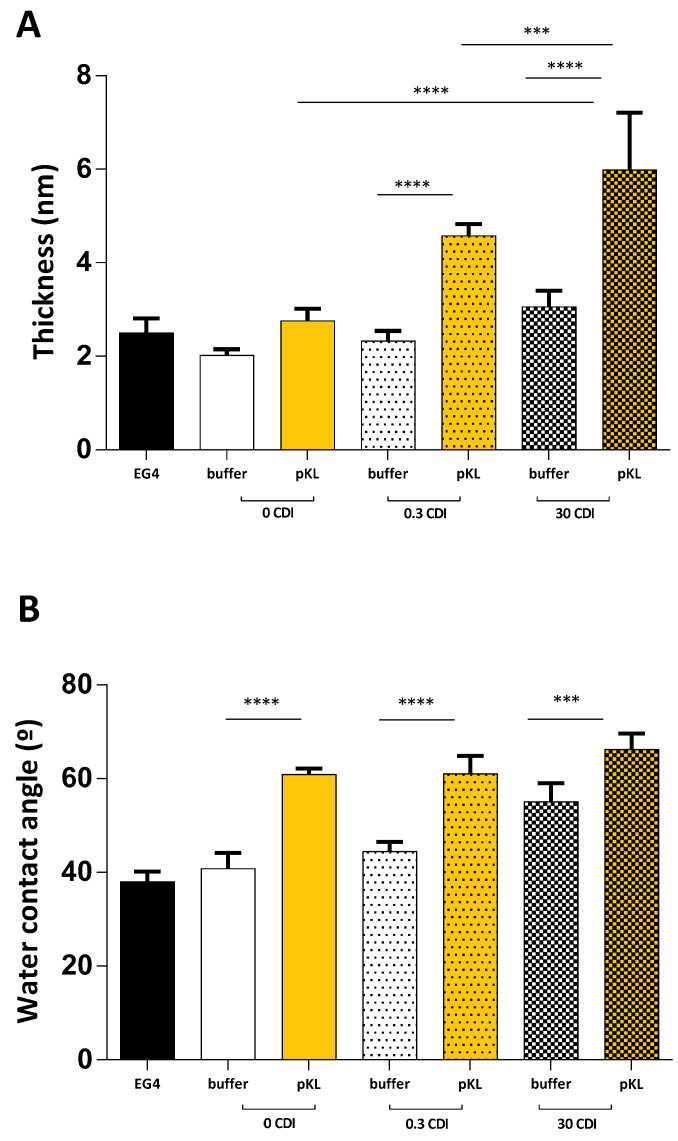
(**A**) **Ellipsometry of pKL-SAMs and buffer-SAMs with increasing CDI concentrations.** Data are expressed as average ± STDEV of three replicates per condition. Statistically significant differences: *** *p* < 0.001; **** *p* < 0.0001. (**B**) **Water contact angle of pKL-SAMs and buffer-SAMs with increasing CDI concentrations**. Data are expressed as average ± STDEV of three replicates per condition. Statistically significant differences: *** *p* < 0.001; **** *p* < 0.0001.

**Figure 2 molecules-28-03422-f002:**
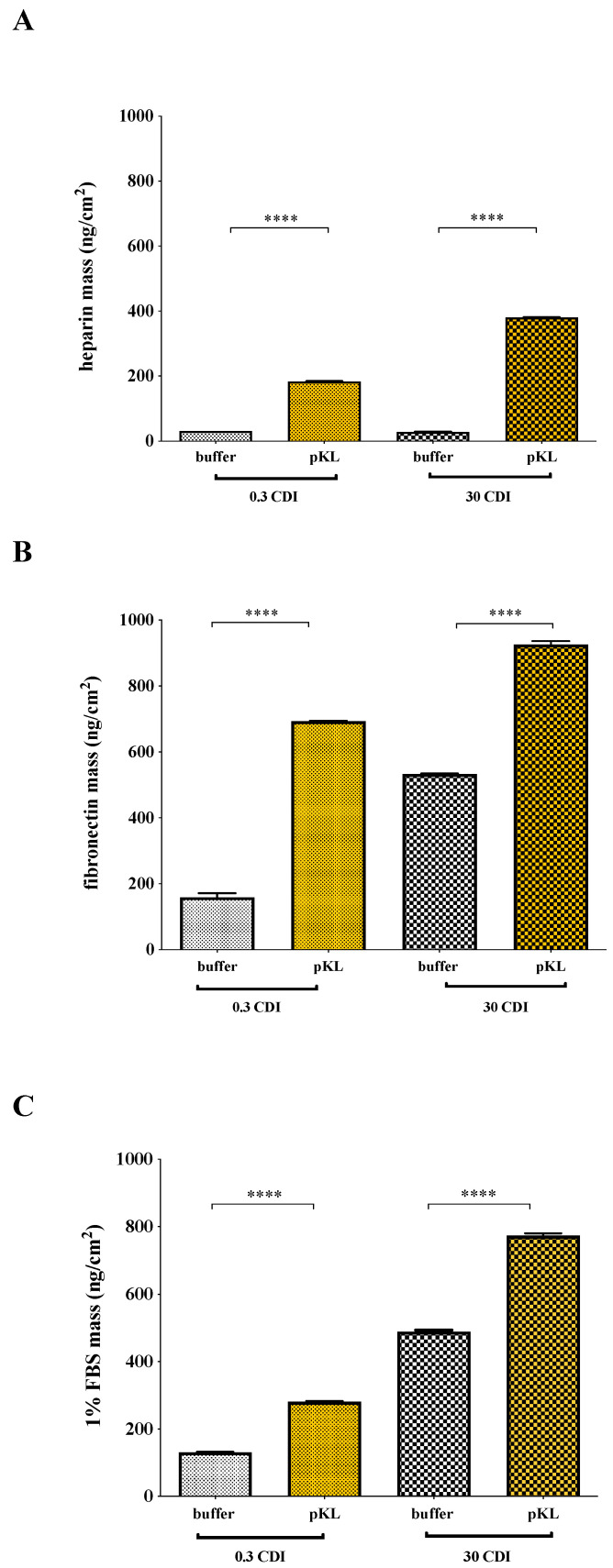
**Protein adsorption at the surfaces as quantified by QCM-D**. QCM-D results for protein adsorption on pKL-SAMs or in buffer-SAMs. (**A**) heparin [0.1 mg/mL]; (**B**) fibronectin [40 μg/mL]; (**C**) 1% (*v*/*v*) FBS solution. Data are expressed as average ± STDEV of three replicates per condition. Statistically significant difference (**** *p* < 0.0001).

**Figure 3 molecules-28-03422-f003:**
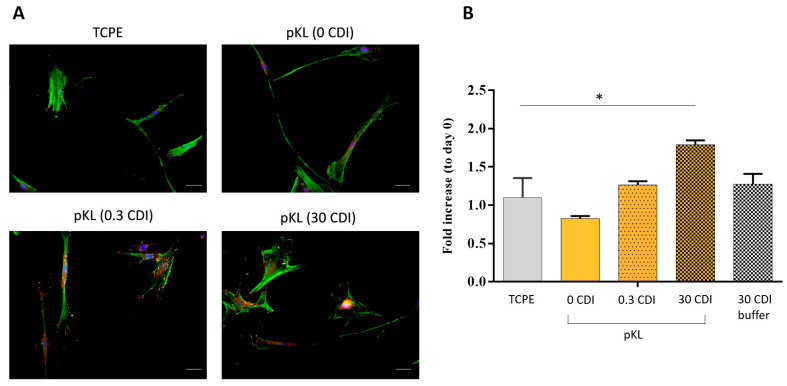
(**A**) Cytoskeleton organization of hMSC 24 h after seeding on TCPE control and pKL-SAMs after activation with different concentrations of CDI. Green: Actin filaments; red: vinculin; blue: nuclei. Scale bar: 50 μm. (**B**) Cells metabolic activity 1 day after seeding, as index of cell adhesion. Data are expressed as average ± STDEV of three replicates per condition. * Statistically significant difference compared to TCPE surface (* *p* < 0.05).

**Figure 4 molecules-28-03422-f004:**
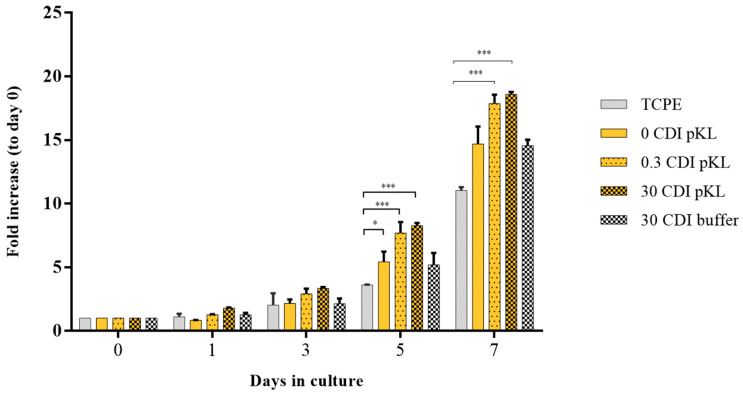
hMSC metabolic activity on pKL-SAMs and buffer-SAMs. Cells metabolic activity during 7 days in culture as index of cell growth. Data are expressed as average ± STDEV of three replicates per condition. * Statistically significant difference as compared to TCPE surface (* *p* < 0.05; *** *p* < 0.001).

## Data Availability

Data presented in this study are contained in the present article and in [24].

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
