# Peer review of "Enhancement of hMSC In Vitro Proliferation by Surface Immobilization of a Heparin-Binding Peptide"

_molecules, 2023, doi:10.3390/molecules28083422_

Round 1

Reviewer 1 Report

Reviews attached

Reviewer 2 Report

The paper entitled “Enhancement of hMSC in vitro Proliferation by Surface 2 Immobilization of a Heparin-Binding Peptide” has a crucial topic and  I think the experiments were very well conducted but the manuscript suffers to be too verbose and often not focused. I think it should be thinned out deeply.

In detail

in the abstract no experimental details should be reported instead, the scientific field the results achieved and the innovation/scientific message of the study.

 scheme 1 is very common for surface functionalization and can be shifted to SI

Lines 228-229, 239-240 rewrite! it is unclear

Results:experimental detail as buffer, reagent concentration and composition should be avoided here in the description of experimental results ma only reported in mat and methods section

Where the authors write more elongated, more formation can they be more quantitative in a comparative way?

The discussion is too long and not focused 

Reviewer 3 Report

The reviewed manuscript written by Cimino et. al presents the pKL peptide-covered surfaces which improve hMSC culture in vitro. The manuscript presents very interesting and worth-reading results and in my opinion is in the journal’s fields of interest. However, I have some reservations thus my suggestion is a major revision. Specific comments are listed below:

  1. Line 45 “Cells sit and organize”. Missed term
  2. Materials and Method section. Could the Authors give more details about pKL synthesis? Any cleaning procedure was applied?
  3. QCM data. Obtained results will be much more clear if the Authors present them as dependencies of adsorbed mass/dissipation as the function of time. 
  4. Line 178 suggests that the viscoelastic properties of obtained layers were analyzed. Where in the Discussion section this subject is analyzed?
  5. Were the stabilities of SAMs, and SAMs with pKL layers on time checked? Cell culture lasts 24 h or longer. Are the Authors sure that during that time SAM/polypeptide surfaces are stable? The Reviewer suggests performing a long-lasting stability test. Additional AFM imaging of substrates stability in time could give more info.
  6. Layer thicknesses together with the small size of pKL (8.5 kDa) suggest polypeptide aggregation on SAMs layers. AFM images would help solve this question.
  7. Adsorbed masses of heparin/fibronectin indicate the formation of a multi-layered/aggregated system. Parallel SPR or OWLS experiments should be conducted to detect the “dry” mass of adsorbed polysaccharide/protein. These layers are probably highly hydrated, too.
  8. The conformation state of fibronectin is also a very important issue. Adsorption in end-on/side-on conformation strongly influences obtained mass. Again AFM imaging could help in this matter.
  9. Lines 399-406. It is a really interesting statement. The described process looks like a cell-built perpetuum mobile. Did the Authors attempt to measure the concentration of the produced growth factors?

Round 2

Reviewer 1 Report

The publication has been correctly revised according to the reviewer's suggestion. There are only a few minor omissions, which should be corrected before publishing:

-line 35: (www.clinicaltrials.gov) should be removed from the text,

-please put spaces where they are needed, e.g. line 35 (results (www.clinicaltrials.gov)[2-5]; line 167 (from[26].), etc.

-lines 240-247- please add a short comment according to the hydrophobicity of the obtained coatings (the definition of the hydrophilic surface with reference, why they are hydrophilic, etc.

-line 256- o” (in „fibronectin”) without Bold

As I mentioned before, these omissions are minor. After improving, there is not necessary to send the corrected paper back to the reviewer. The manuscript can be published in Molecules.

Author Response

Response to Reviewer 1 Comments

Major comments:

The publication has been correctly revised according to the reviewer's suggestion. There are only a few minor omissions, which should be corrected before publishing:

-line 35: (www.clinicaltrials.gov) should be removed from the text,

-please put spaces where they are needed, e.g. line 35 (results (www.clinicaltrials.gov)[2-5]; line 167 (from[26].), etc.

Changes were performed according to the Reviewer comments

-lines 240-247- please add a short comment according to the hydrophobicity of the obtained coatings (the definition of the hydrophilic surface with reference, why they are hydrophilic, etc.

We thank the reviewer for the suggestion.

A sentence has been added to the text together with its reference

“Contact angle measurement is a qualitative way to assess surface wettability. It is based on the evaluation of the intermolecular interactions between the surface and a small drop of water when the drop meets the surface. Water contact angles of hydrophilic surfaces are typically smaller than 90° while hydrophobic ones have contact angels greater than 90° [32].”

[32]      Mittal, K.L., Contact Angle, Wettability and Adhesion. Vol. 6. 2009: CRC Press.

-line 256- „o” (in „fibronectin”) without Bold

Change was performed according to the Reviewer comments

As I mentioned before, these omissions are minor. After improving, there is not necessary to send the corrected paper back to the reviewer. The manuscript can be published in Molecules.

Reviewer 2 Report

Rge authors satisfied all my previous issues

Author Response

Response to Reviewer 2 Comments

Major comments:

Rge authors satisfied all my previous issues

We thank the reviewer for the suggestions that help us to improve the manuscript

Reviewer 3 Report

very interesting QCM data. It's worth analyzing them in a separate paper.

Author Response

Response to Reviewer 3 Comments

Major comments:

very interesting QCM data. It's worth analysing them in a separate paper.

We thank the reviewer for the suggestions that help us to improve the manuscript.